# Satisfaction and associated factors towards inpatient health care services among adult patients at Pawie General Hospital, West Ethiopia

Tesgera Begize Aga[1], Yohannes Mulu Ferede[1], Enyew Getaneh Mekonen[2]*

1 Department of Medical Nursing, School of Nursing, College of Medicine and Health Sciences, University of Gondar, Gondar, Ethiopia, 2 Department of Surgical Nursing, School of Nursing, College of Medicine and Health Sciences, University of Gondar, Gondar, Ethiopia

* enyewgetaneh111@gmail.com

## Abstract

### Introduction

Improving the quality of services is the primary goal of the Ethiopia reform program to satisfy patients. Patient satisfaction is an attitude resulting from a person's general orientation towards a total experience of health care. According to world health organization consumer satisfaction is playing an increasingly important role in the quality of care reforms and health-care delivery more generally.

### Objective

To assess patient's satisfaction and associated factors with health care services among admitted patients in Pawie General Hospital, Benishangul Gumuze Region, West Ethiopia, 2020

### Methods

Institution based cross-sectional study was conducted among adult patients admitted to Pawie General Hospital. A systematic random sampling technique was employed to recruit 334 participants and a structured interviewer-administered questionnaire was used to collect data. Data were entered into Epi Data version 3.1, analyzed using SPSS version 23, and presented in tables and graphs. Bivariable and multivariable logistic regressions were computed to identify factors associated with patient satisfaction. P-values < 0.05 and adjusted odds ratios were used to declare the significance and strength of the association.

### Result

The overall patient's satisfaction towards inpatient health care services at Pawie General Hospital was 60.8% with 95% CI (55.4, 65.9). Factors like admission ward [AOR = 2.60; 95% CI (1.34, 5.03)] and privacy [AOR = 12.5; 95% C I (2.89, 54.1)] were significantly associated with patient's satisfaction.

**Data Availability Statement:** All relevant data are within the manuscript and its Supporting Information files.

**Funding:** The authors received no specific funding for this work.

**Competing interests:** The authors have declared that no competing interests exist.

**Abbreviations:** AOR, Adjusted Odds Ratio; CRC, Compassionate Respectful and Caring; FMOH, Federal Ministry of Health; SPSS, Statistical Package for the Social Sciences; WHO, World Health Organization.

## Conclusion

The satisfaction level of patients admitted to Pawie General Hospitals was low. Admission ward and perceived privacy assured were factors significantly associated with patient satisfaction among patients admitted to Pawie General Hospital. The hospital administration system is better to work together to fill the gaps identified and improve the level of patient satisfaction.

## Introduction

Patient satisfaction is an essential component of client-centered care and important quality of care indicator. It affects the accreditation of health institutions they need to prove whether they meet a general standard of quality care or not [1, 2]. Patient satisfaction during hospitalization represents a balance between the patient's perception and expectation of their health care services [3]. According to world health organization (WHO) consumer satisfaction is playing an increasingly important role in the quality of care reforms and health-care delivery more generally. Yet, studies were done by WHO showed that 46% of all the patients that visit different health care facilities, like Hospitals, are dissatisfied with the service they received [4]. In general, hospitals and healthcare systems that invest in citizens' evaluation and patients' assessment programs, are expected to acquire valuable information to perform important transformational changes reforms in their healthcare services [5]. Dissatisfaction increases anxiety and irritability in patients has resulted in delayed recovery time and more beds of the hospital will be occupied by increasing the length of hospitalization and costs of treatment [6].

In developed countries, patients' satisfaction surveys have improved the quality of health care delivery and have become a mandatory issue in almost all their hospitals [7, 8]. Studies have indicated that services marketing has a greater effect on patient behavior than the costs they have to endure in hospitals. As a result, services marketing can describe a positive image of the Hospital, which in turn encourages patients to remain more loyal and refer the facility to others [9]. A descriptive study conducted in Uganda showed that the majority of the patients prefer to attend private facilities than public facilities due to poor satisfaction with health care services [10].

Patients who are not satisfied with a service may have worse outcomes because, they miss appointments, leave against the advice, or fail to follow on treatment plans [11]. The satisfaction level of the patients is inadequate in some studies done in Ethiopia, so this demands to take the further assessment and take action on the identified problems to improve the services delivered to the patients [12].

The solutions tried for patient satisfaction in the health care system have given the attention for acceptability and preference of patients in the hospital, by making the environment comfortable [3]. Federal Ministry of Health (FMOH) has developed training materials on Compassionate Respectful and Caring (CRC) and provided for health care professionals, strengthening the management capacity of health facilities to improve the quality of health services to satisfy the community's health need [13]. However, there is no study conducted to assess the admitted patient's satisfaction on health care services in Pawie General Hospital. Therefore, this study aimed to assess patient satisfaction and associated factors with health care services among admitted patients in Pawie General Hospital.

## Methods and materials

### Study design and period

Institutional based cross-sectional study was conducted from March 4 to April 14, 2020.

## Study setting

The study was conducted at Pawie General Hospital in Metekele Zone, Pawie Woreda of Benishangul Gumuze Regional State. The hospital has five wards (Medical, Surgical, Pediatrics, Obstetrics, and Gynecology wards) that serve the population of the Zone and nearby Amhara region. The capital city of Metekele Zone is Gelgele Beles. It has a latitude and longitude of 10° 25' 36" N / 35° 43' 11" E. Pawie district is found 575km far from Addis Ababa, the capital city of Ethiopia and 401 km from Asossa, the capital city of Benishangul Gumuze Regional State. Based on the 2007 Census conducted by Ethiopia, this Zone has a total population of 276,367, of whom 139,119 are men and 137,248 women. 37,615 or 13.61% of the population are urban inhabitants. Metekele Zone has one General hospital, two primary Hospitals and 19 public health centers, and seven private clinics. According to the 2019 Pawie General Hospital human resource administration report, the hospital is expected to serve more than 468,353 peoples of the Metekele Zone and part of Chagini and Jawie Woreda of Awie Zone in Amhara region. The hospital gives different inpatient and outpatient services, including TB, HIV care service which was started in 2003, operation services.

## Study participants

All admitted adult patients who have >24 hours of admission at Pawie General Hospital during the study period were included in the study. Those admitted patients who are seriously ill or unconscious were excluded.

## Sample size determination and sampling procedure

The sample size was calculated using Epi info version 7 stat cal by taking the estimated proportion of inpatient satisfaction 79.7% [14], a confidence level of 95%, and margin of error 5% for satisfaction. While for factors associated with patient satisfaction assuming comparative cross-sectional; unexposed: exposed (1:1). The final sample size was 334 after adding a 10% non-response rate. A systematic random sampling technique was employed to select study participants in the study area by using their inpatient registration. The total sample size was distributed to the wards (Medical, Surgical, and Gynecology/Obstetrics ward) in the hospital using proportional allocation based on the average number of participants in previous past months. The first patient to be included in the study was selected by the lottery method, (K = N/n, where; K = the interval, N = total population, and n = sample size) and then, every 2 participants were interviewed.

## Data collection instruments and procedures

Data were collected using the structured interviewer-administered questionnaire adapted from different literature after discharge from wards [3, 14, 15]. Fifty-seven questionnaires which had three parts were used: part I- socio-demographic characteristics of the patients (9 items), part II-Hospitalization and personal related condition (11 items), and part III- patient satisfaction (36 items) of the respondents and their level of satisfaction with the hospital services. Each tool used to measure the satisfaction rate involved 5-point Likert scale response (1 = very dissatisfied, 2 = dissatisfied, 3 = neutral, 4 satisfied, 5 = very satisfied). Data were collected with the help of three BSc nurses and two MSc Nurse Supervisors.

## Data processing and analysis

The collected data were checked for completeness and consistency and then coded and entered into Epi Data version 3.1 and exported to SPSS version 23 for analysis. Descriptive statistics

like frequency, percentage, graphs, and tables were used for data presentation. By summing up the response of 36 satisfaction questions those who scored points more than or equal to the mean score were categorized as satisfied and those patients who scored less than the mean score were categorized as dissatisfied. The bivariable logistic regression model was used to test if there is an association between a dependent variable and each independent variable. Factors statistically significant at a p-value of 0.2 and less at bivariate logistic regression were taken to multivariable logistic regression. Variables having a p-value < 0.05 in the multivariable logistic regression were considered as factors affecting outcome variables. Finally, the adjusted odds ratio was used to determine the strength of the association between a dependent variable and independent variables.

### Data quality control

The data collection instruments were reviewed by four experts (two clinical nurses and two nurse academics). The tools were also tested with a pretest by taking 5% of the sample size before the actual data collection time at Asossa General Hospital with the same level but different from the study hospital to ensure the comprehensibility, and understandability of the tools. Some redundancy and vague sentences were corrected. The one-day training was given for data collectors before data collection. Regular supervision, spot-checking, and reviewing the completed questionnaire was carried out daily by the principal investigator and supervisors. The reliability of the tools was checked (Cronbach's alpha > 0.81).

### Ethical consideration

Before conducting the study, ethical clearance was obtained from the school of Nursing on behalf of the Institutional Review Board of the University of Gondar. Upon this clearance, additional written permission to conduct the study on patients was obtained from the Chief Executive Officer of the Hospital. Written informed consent was obtained from each participant and they are also informed that they have the right to withdraw from the study at any point in time. Even though it is patients' interview, the confidentiality of information and privacy of the patients was maintained by avoiding recording of the patient's name on the questionnaire and keeping the data anonymous, and also the recorded information was not used other than the study purpose. The issue of privacy and confidentiality was strictly maintained.

## Results

### Socio-demographic characteristics of the respondents

A total of 334 patients were involved in this study with a response rate of 100%. The mean age of the respondents was 30.6 ±10.6 years (standard deviation) and more than one third (36.2%) of them were found in the age range of 18–25 years. More than two-thirds (69.2%) of the respondents were females and nearly three-fourth (73.4%) of them were orthodox Christian in religion. One hundred and fifteen (34.4%) of the respondents had no formal education and more than half (54.2%) of them lived in an urban area. The majority of the respondents (83.2%) were married and more than half (51.8%) of them were farmers. Ninety-nine (29.6%) of the study participants earned a monthly income of 500–1500 Ethiopian Birr (Table 1).

### Patient-related factors

The majority (90.7%) of the respondents were admitted to the hospital through emergency and nearly two-thirds (63.2%) of them stayed less than 3 days. Two hundred and five

**Table 1. Sociodemographic characteristics of the respondents at Pawie General Hospital Benishangul Gumuze Region, West Ethiopia, 2020 (n = 334).**

| Variables | Category | Frequency | Percent (100%) |
|---|---|---|---|
| Sex | Male | 103 | 30.8 |
| | Female | 231 | 69.2 |
| Age | 18–25 years | 121 | 36.2 |
| | 26–33 years | 115 | 34.4 |
| | 34–41 years | 55 | 16.5 |
| | > = 42 years | 43 | 12.9 |
| Marital status | Single | 44 | 13.2 |
| | Married | 278 | 83.2 |
| | Divorced | 8 | 2.4 |
| | Widowed | 4 | 1.2 |
| Educational status | No formal education | 115 | 34.4 |
| | Primary (1–8) | 113 | 33.8 |
| | Secondary (9–12) | 43 | 12.9 |
| | Diploma & above | 63 | 18.9 |
| Occupation | Govt. employee | 49 | 14.6 |
| | Marchant | 29 | 8.7 |
| | Farmer | 173 | 51.8 |
| | Non employer | 7 | 2.1 |
| | Student | 48 | 14.4 |
| | Housewife | 28 | 8.4 |
| Religion | Orthodox | 245 | 73.4 |
| | Muslim | 59 | 17.6 |
| | Protestant | 29 | 8.7 |
| | Catholic | 1 | 0.3 |
| Residence | Urban | 181 | 54.2 |
| | Rural | 153 | 45.8 |
| Monthly income | 500–1500 ETB | 99 | 29.6 |
| | 1501–2500 ETB | 90 | 26.9 |
| | 2501–3500 ETB | 71 | 21.3 |
| | >3500 ETB | 74 | 22.2 |

ETB = Ethiopian Birr

(61.4%) of the respondents were admitted to the hospital for the first time and more than three fourth (79.3%) of them were admitted for an acute case. One hundred fifty-nine (47.6%) of the respondents were admitted in the evening time and only twenty-seven (8.1%) of them had comorbid with other diseases. More than half (56.9%) of the patients had got the services with payment and only thirty-nine (11.7%) were getting the services with privacy assured (Table 2).

## Hospital related factors

More than half (56%) of the respondents were not satisfied with the cleanliness of the toilet. The majority (98.2%), (97.9%), and (97.3%) of the respondents were satisfied with the quietness of the room for rest, access to requested laboratory tests, and room light and ventilation respectively. Three hundred and thirty (98.8%) of the respondents were satisfied with the coherence of the service units in the hospital. Majority (85.3%) of the study participants were satisfied with the condition of the food. More than three-fourths (76.6%) of the respondents

**Table 2. Patient-related factors of inpatient services at Pawie General Hospital Benishangul Gumuze Region, West Ethiopia, 2020 (n = 334).**

| Variables | Category | Frequency | Percent (100%) |
|---|---|---|---|
| Experience of admission | New visit | 205 | 61.4 |
|  | Repeat visit | 129 | 38.6 |
| Admission mode | Emergency | 303 | 90.7 |
|  | Planned | 31 | 9.3 |
| Comorbidity | Yes | 27 | 8.1 |
|  | No | 307 | 91.9 |
| Duration of Hospital stay | 1–3 days | 211 | 63.2 |
|  | 4–7 days | 117 | 35.0 |
|  | >8 days | 6 | 1.8 |
| Payment for services | With payment | 190 | 56.9 |
|  | Free | 144 | 43.1 |
| Time of Hospitalization | Morning | 143 | 42.8 |
|  | Evening | 159 | 47.6 |
|  | Night | 32 | 9.6 |
| Privacy assured during the examination | Yes | 39 | 11.7 |
|  | No | 295 | 88.3 |
| Waiting time for admission | <1 day | 319 | 95.5 |
|  | 1–3 days | 9 | 2.7 |
|  | >3 days | 6 | 1.8 |

were satisfied with the availability of drinking water. Only 3.6% the study participants were dissatisfied with accommodation of the room. More than three-fourths (76.6%) of the respondents were satisfied with the clean and comfortableness of the room (Fig 1).

## Overall patient satisfaction

The overall patient's satisfaction towards inpatient health care services at Pawie General Hospital was 60.8% with 95% CI (55.4, 65.9) (Fig 2).

## Factors associated with patient's satisfaction

Variables like sex, occupation, monthly income, admission ward, service fee, and privacy were significantly associated with the outcome variable in the bivariable analysis. In multivariable logistic regression analysis, factors significantly associated with patient satisfaction were the admission ward and privacy. The odds of satisfaction for patients who were admitted to the surgical ward was 2.6 times higher than those patients admitted to other wards [AOR = 2.60; 95% CI (1.34, 5.03)]. Participants who report their feeling on ways privacy was assured were 12.5 times more likely to be satisfied than participants in whom measures were not taken to assure privacy [AOR = 12.5; 95% CI (2.89, 54.1)] (Table 3).

## Discussion

In this study, the overall patients' satisfaction was 60.8% with 95% CI (55.4, 65.9). This finding was relatively consistent with a study conducted in Iran (58%) and Jimma University Specialized Hospital, Ethiopia (61.9) [16, 17]. However, the finding of this study was lower than studies conducted in Australia (69.3) Nigeria (96.1%) and China (80%) [3, 15, 18]. The possible justification for this difference might be the difference in the methodology used. The study conducted in China used a mixed-methods approach. But the current study used a quantitative

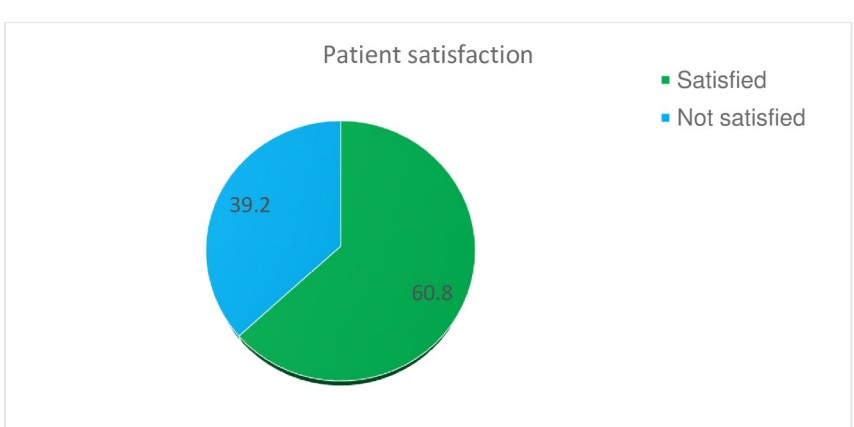

**Fig 1. Satisfaction with hospital-related factors among patients admitted to Pawie General Hospital Benishangul Gumuze Region, West Ethiopia, 2020 (n = 334).**

**Fig 2. Overall satisfaction of patients admitted to Pawie General Hospital Benishangul Gumuze Region, West Ethiopia, 2020 (n = 334).**

**Table 3. Bivariable and multi-variable analysis of factors associated with satisfaction among patients admitted to Pawie General Hospital, West Ethiopia, 2020.**

| Variables | | Satisfaction Status | | OR with 95% C I | | |
|---|---|---|---|---|---|---|
| | | Satisfied | Dissatisfied | Crude | Adjusted | P-value |
| Sex | Male | 49 | 53 | 1 | 1 | |
| | Female | 154 | 78 | 0.47 (0.29, 0.75) | 0.86 (0.48, 1.54) | 0.622 |
| Occupation Government employee | | 26 | 24 | 1 | 1 | |
| Merchant | | 16 | 13 | 0.88 (0.06, 0.99) | 0.28 (0.08, 1.04) | 0.057 |
| Farmer | | 104 | 68 | 0.71 (0.06, 0.94) | 0.29 (0.07, 1.20) | 0.088 |
| Jobless | | 4 | 3 | 0.81 (0.09, 0.98) | 0.49 (0.15, 1.61) | 0.242 |
| Students | | 29 | 19 | 0.71 (0.04, 1.39) | 0.30 (0.42, 2.12) | 0.226 |
| Others | | 24 | 4 | 0.18 (0.08, 0.85) | 0.40 (0.11, 1.48) | 0.170 |
| Monthly income | | | | | | |
| 500–1500 ETB | | 67 | 32 | 1 | 1 | |
| 1501–2500 ETB | | 55 | 35 | 1.33 (1.12, 3.11) | 1.68 (0.79, 3.59) | 0.182 |
| 2501–3500 ETB | | 45 | 26 | 1.21 (0.89, 3.09) | 1.14 (0.54, 2.36) | 0.732 |
| >3500 ETB | | 36 | 38 | 2.21 (0.94, 4.55) | 1.57 (0.75, 3.28) | 0.236 |
| Admission ward Gynecology/Obstetrics | | 76 | 17 | 1 | | |
| Surgical Ward | | 61 | 67 | 4.91 (2.67, 7.07) | **2.60 (1.34, 5.03)** * | 0.004 |
| Medical ward | | 66 | 47 | 0.65 (0.39, 1.08) | 0.57 (0.34, 1.96) | 0.079 |
| Service fee | | | | | | |
| With payment | | 96 | 95 | 1 | 1 | |
| Free | | 107 | 36 | 0.34 (0.21, 0.54) | 0.63 (0.33, 1.20) | 0.164 |
| Privacy | Assured | 36 | 2 | 13.9 (3.29, 58.8) | **12.5 (2.89, 54.1)** * | 0.001 |
| | Not assured | 167 | 129 | 1 | 1 | |

* Statistically significant at p-value <0.05

approach only. Detail information can have obtained through a qualitative approach. The study conducted in Nigeria used an observational design. The observational study enables the researcher to collect the right information that patients may report in the wrong ways. It was also lower than studies conducted in Black Lion (90.1%) and Felege Hiwot Referral Hospital, Ethiopia (74.9%) [19, 20]. This might be due to the difference in study participants and the service provided to patients. The Felege Hiwot Referral Hospital study was conducted to assess women's satisfaction with childbirth care. Exempted from any payment for childbirth care services, and increased government concern for maternal health service may increase the satisfaction status. The Black Lion Referral Hospital study was conducted to assess adult patients' satisfaction with nursing care provided. Whereas the current study was conducted to assess adult patients' satisfaction with all inpatient services provided. Assessing the patients' satisfaction with a specific service provided (nursing care) may increase the satisfaction status of patients. On the other hand, it was higher than studies conducted in the Gambia (36.8%) and Arba Minch hospital, Gamo Gofa, Ethiopia (40.9%) [17, 21]. The plausible justification for this difference might be differences in the study setting, characteristics of study participants, and tools used. The Gambia study was conducted in a referral and teaching hospital while the current study was conducted in a general hospital. The expectation of patients admitted to a referral hospital might be high and their satisfaction status becomes low as compared with patients admitted to a general hospital.

In this study, patients admitted to the surgical ward were 2.6 times more likely to be satisfied than patients admitted to other wards. This finding was supported by studies conducted in Jimma University Specialized Hospital and Public Hospitals of Amhara Region [3, 22]. This

might be due to the condition and expectation of patients admitted to the medical ward and Gynecology/ Obstetrics ward. Patients admitted to the medical ward are mostly diagnosed with more severe conditions, poorer prognosis, and being greatly exposed to stressful and anxious situations. As a result, their perception level might be influenced by the hallo effect (a cognitive bias) of these factors which made their satisfaction level low. Similarly, participants who report their feeling on ways privacy was assured were 12.5 times more likely to be satisfied than participants in whom measures were not taken to assure privacy. This finding was in line with studies conducted in Amhara Region Public Hospitals and Asella Hospital of Oromia Region [23, 24]. This might be due to a lack of privacy that makes communication difficult between patients and healthcare professionals, mainly when they discuss sensitive conditions. Misdiagnosis and ineffective treatments for patients may result if the patients' privacy is not assured. Finally, the patient's trust may have eroded which makes it difficult to build a good patient- healthcare professional relationship. As a result, the patient's satisfaction level may decrease if privacy is not assured during care provision.

## Strength of the study

The response rate was high. Since the interview was made after discharge at the exit patients are free to respond.

## Limitation of the study

The cause/effect and the temporal relationship could not be established due to the cross-sectional nature of the study.

There might be an introduction of confirmation bias since data were collected by health professionals.

## Conclusion

The satisfaction of patients admitted to Pawie General Hospital was low. Admission ward and privacy were factors associated with patient satisfaction among patients admitted to Pawie General Hospital. The hospital administration system is better to work together to fill the gaps identified and improve the level of patient satisfaction.

## Supporting information

**S1 Data.**
(XLS)

## Acknowledgments

The authors are grateful to the University of Gondar, Pawie General Hospital director, data collectors, and study participants.

## Author Contributions

**Conceptualization:** Tesgera Begize Aga, Yohannes Mulu Ferede, Enyew Getaneh Mekonen.

**Data curation:** Tesgera Begize Aga.

**Formal analysis:** Tesgera Begize Aga.

**Methodology:** Yohannes Mulu Ferede, Enyew Getaneh Mekonen.

**Supervision:** Yohannes Mulu Ferede, Enyew Getaneh Mekonen.

**Writing – original draft:** Tesgera Begize Aga, Enyew Getaneh Mekonen.

**Writing – review & editing:** Yohannes Mulu Ferede, Enyew Getaneh Mekonen.

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
