## [Decision Letter · Decision Letter 0]

4 Dec 2020

PONE-D-20-24310

Satisfaction and associated factors towards inpatient health care services among adult patients at Pawie General Hospital, Benishangul Gumuze Regional State, West Ethiopia

PLOS ONE

Dear Authors,

Thank you for submitting your manuscript to PLOS ONE. After careful consideration, we feel that it has merit but does not fully meet PLOS ONE’s publication criteria as it currently stands. Therefore, we invite you to submit a revised version of the manuscript that addresses the points raised during the review process.

Please submit your revised manuscript within three weeks. If you will need more time than this to complete your revisions, please reply to this message or contact the journal office at plosone@plos.org. Please include the following items when submitting your revised manuscript:

We look forward to receiving your revised manuscript.

Kind regards,

Bishwajit Ghose, PhD.

Academic Editor

PLOS ONE

Journal Requirements:

2. Please address the following:

- Please include additional information regarding the survey or questionnaire used in the study and ensure that you have provided sufficient details that others could replicate the analyses. For instance, if you developed a questionnaire as part of this study and it is not under a copyright more restrictive than CC-BY, please include a copy, in both the original language and English, as Supporting Information.

- Please ensure you have thoroughly discussed any potential limitations of this study within the Discussion section, including the potential introduction of bias during data collection.

3. Please modify the title to ensure that it is meeting PLOS’ guidelines (https://journals.plos.org/plosone/s/submission-guidelines#loc-title). In particular, the title should be "specific, descriptive, concise, and comprehensible to readers outside the field" and in this case we suggest it could be shortened, for example: "Satisfaction and associated factors towards inpatient health care services among adult patients at Pawie General Hospital, West Ethiopia".

4.  Please provide additional details regarding participant consent. In the ethics statement in the Methods and online submission information, please ensure that you have specified what type you obtained (for instance, written or verbal, and if verbal, how it was documented and witnessed). If your study included minors, state whether you obtained consent from parents or guardians. If the need for consent was waived by the ethics committee, please include this information.

6. We noticed you have some minor occurrence of overlapping text with the following previous publication(s), which needs to be addressed:

https://www.hindawi.com/journals/ogi/2018/2475059/

In your revision ensure you cite all your sources (including your own works), and quote or rephrase any duplicated text outside the methods section. Further consideration is dependent on these concerns being addressed.

Reviewers' comments:

Reviewer's Responses to Questions

**Comments to the Author**

1. Is the manuscript technically sound, and do the data support the conclusions?

Reviewer #1: Yes

Reviewer #2: Yes

2. Has the statistical analysis been performed appropriately and rigorously? 

Reviewer #1: Yes

Reviewer #2: Yes

3. Have the authors made all data underlying the findings in their manuscript fully available?

Reviewer #1: Yes

Reviewer #2: Yes

4. Is the manuscript presented in an intelligible fashion and written in standard English?

Reviewer #1: Yes

Reviewer #2: Yes

5. Review Comments to the Author

Reviewer #1: The authors studied the patients's satisfaction of the hospital.

The data seems to be a good reference for other hospital operation.

Although the important results are described in the Tables, it is difficult to understand the results.

I suggest that the author used more comprehensive chart and figures.

Reviewer #2: Paper suggests and supports the unusual importance of patients’ satisfaction studies. Patients’ satisfaction is independent of the type of medical facility, private or public. According to presented data, the satisfaction level to some extent is the measure of therapy/treatment efficacy. Human expectations, regrading medical healthcare are universal and independent of geographical location.

6. PLOS authors have the option to publish the peer review history of their article (what does this mean?). If published, this will include your full peer review and any attached files.

Reviewer #1: No

Reviewer #2: No

---

## [Author Response · Author response to Decision Letter 0]

6 Jan 2021

We tried to incorporate and address all the comments and suggestions forwarded by an academic editor and reviewers.

---

## [Decision Letter · Decision Letter 1]

15 Mar 2021

Satisfaction and associated factors towards inpatient health care services among adult patients at Pawie General Hospital, West Ethiopia

PONE-D-20-24310R1

Dear Authors,

We’re pleased to inform you that your manuscript has been judged scientifically suitable for publication and will be formally accepted for publication once it meets all outstanding technical requirements.

Kind regards,

Bishwajit Ghose, PhD.

Academic Editor

PLOS ONE

Additional Editor Comments (optional):

Reviewers' comments:

Reviewer's Responses to Questions

**Comments to the Author**

1. If the authors have adequately addressed your comments raised in a previous round of review and you feel that this manuscript is now acceptable for publication, you may indicate that here to bypass the “Comments to the Author” section, enter your conflict of interest statement in the “Confidential to Editor” section, and submit your "Accept" recommendation.

Reviewer #1: All comments have been addressed

2. Is the manuscript technically sound, and do the data support the conclusions?

Reviewer #1: Yes

3. Has the statistical analysis been performed appropriately and rigorously? 

Reviewer #1: Yes

4. Have the authors made all data underlying the findings in their manuscript fully available?

Reviewer #1: Yes

5. Is the manuscript presented in an intelligible fashion and written in standard English?

Reviewer #1: Yes

6. Review Comments to the Author

Reviewer #1: Authors well upgraded the manuscript figures, therefore, I suggest "accept" for publication as it is.

7. PLOS authors have the option to publish the peer review history of their article (what does this mean?). If published, this will include your full peer review and any attached files.

Reviewer #1: No

---

## [Editor Report · Acceptance letter]

30 Mar 2021

PONE-D-20-24310R1 

Satisfaction and associated factors towards inpatient health care services among adult patients at Pawie General Hospital, West Ethiopia 

Dear Dr. Mekonen:

I'm pleased to inform you that your manuscript has been deemed suitable for publication in PLOS ONE. Congratulations! Your manuscript is now with our production department. 

Kind regards, 

on behalf of

Dr. Bishwajit Ghose 

Academic Editor

PLOS ONE